# Informal caregiver support needs and burden: a survey in Lithuania

Ieva Biliunaite ,[1] Evaldas Kazlauskas ,[2] Robbert Sanderman,[3,4] Gerhard Andersson[1,5,6]

¹Department of Behavioural Sciences and Learning, Linköping University, Linköping, Sweden
²Center for Psychotraumatology, Institute of Psychology, Vilnius University, Vilnius, Lithuania
³Department of Health Psychology, University Medical Center Groningen, University of Groningen, Groningen, The Netherlands
⁴Department of Psychology, Health and Technology, University of Twente, Enschede, The Netherlands
⁵Department of Biomedical and Clinical Sciences, Linköping University, Linköping, Sweden
⁶Department of Clinical Neuroscience, Karolinska Institutet, Stockholm, Sweden

**Correspondence to**
Ieva Biliunaite;
ieva.biliunaite@liu.se

## ABSTRACT

**Introduction** A demand for informal care exists worldwide. Lithuania presents an interesting case example where the need for the informal care is increasing, but relatively little research has been conducted documenting caregivers' experiences and needs.

**Objectives** The main objective of this study was to investigate Lithuanian informal caregiver characteristics, support needs and burden. In addition, the impact of the COVID-19 on the caregiver's and care receiver's well-being was investigated.

**Methods** The study was conducted online between May and September 2020. Informal caregivers and individuals with informal caregiving experiences were invited to participate in the survey. The survey questionnaire comprised 38 multiple-choice items including participant demographic characteristics, availability of the support, support needs, well-being and the impact of the COVID-19 pandemic. In addition, caregiver burden was assessed with the 24-item Caregiver Burden Inventory (CBI).

**Results** A total of 226 individuals completed the survey. Most of the participants were women (87.6%). Almost half of the participants (48.7%) were not receiving any support, and a total of 73.9% expressed a need to receive more professional support. Participants were found to experience high burden on the CBI (M=50.21, SD=15.63). Women were found to be significantly more burdened than men (p=0.011). Even though many participants experienced psychological problems (55.8%), only 2.2% were receiving any psychological support. Finally, majority of the participants did not experience any changes in their own (63.7%) or the well-being of their care receiver (68.1%) due to the COVID-19 pandemic.

**Conclusion** Most of the participants were identified as intensive caregivers experiencing a high burden. A majority did not experience changes in their well-being due to COVID-19. We propose several recommendations for increasing accessibility and availability of support for informal caregivers in Lithuania based on the study findings.

## Strengths and limitations of this study

► The study was designed to explore needs of growing, yet scarcely researched population of Lithuanian informal caregivers.
► The study targeted a wide range of informal caregivers, providing care in the context of disability, illness, old age or frailty.
► Taking part in the survey was voluntary and it is likely that the sample is not fully representative.
► Limitations of the self-report data should be considered in interpreting the survey's findings.

increases in the longevity and demand for long-term care, and the limited resources for formal care, it is likely that more people will need to be involved in informal caregiving in the future.[2] It is evident that informal caregivers are not only important for the management of the long-term care, but that they also carry a substantial economic cost[3] and hence, form a backbone of the health and societal care delivery worldwide.[4]

Informal caregiving experience can vary greatly depending on several factors. For example, the motivation to provide care,[5] intensity of the caregiving,[6] caregivers skills,[7] and the symptoms of the care receiver are likely to influence the care.[8] It is known that caregiving can lead to positive experiences, such as personal growth or feelings of closeness and intimacy with the care receiver.[9] At the same time, caregivers also experience worse psychological[10] and physical health[11] than non-carers. In addition, they are at risk of loneliness and social isolation,[12] as well as financial difficulties.[13] Accumulation of these negative outcomes can increase caregiver burden, an experience that is described as a combination of the psychological, physical, social and financial strains.[4] This concerns many women, as they make-up a majority of informal caregivers,[2] and tend to experience worse negative mental

## INTRODUCTION

Informal caregivers are individuals, who despite no training or experience, provide care for significant others such as partners, children, siblings, parents or friends. In Europe, the proportion of individuals involved in some form of informal care ranges between 20% and 44%.[1] Due to the

health outcomes than men informal caregivers.[4] Altogether, caregiving could be described as a complex experience that often put caregivers at risk of worse psychological health. Providing caregivers with effective support could help to prevent negative outcomes and improve their quality of life, and also improve quality of care for the care receiver.[14]

Several studies have been conducted over the last years investigating informal caregiver support needs. Some of the more commonly observed needs are the need for information and education in the care provision,[15] a need for better collaboration with healthcare professionals,[16] flexible work arrangements,[17] a need for social recognition,[18] as well as the availability of professional support.[16] Despite the commonalities of caregiver needs, research show variation in the caregiver well-being across countries.[19] That is, caregiver needs vary based on the country of residence and the specific cultural and socioeconomic background. For this reason, research on country-specific needs of informal caregivers that considers demographic, cultural and economical influences is needed.[18]

In this study, we focused on Lithuanian informal caregivers. As in many other European countries, in Lithuania demand for the informal care is increasing, while the availability of such care is decreasing. This problem is even more evident in Lithuania, were due to mass emigration and one of the fastest ageing populations in Europe, informal care resources are shrinking rapidly.[20] Lithuanian constitution states that it is the duty of the children to take care of their parents.[21] According to the previous findings,[20] more than half of the middle-aged respondents agree with this statement and would prefer to receive informal care themselves. Despite that, social policy measures were previously found to be inadequate in meeting the expectations for the informal care as well as allowing existing informal caregivers to balance their personal, work and caregiving-related duties.[20] The available literature regarding Lithuanian informal caregivers' needs is very limited and is mostly based on small scale qualitative findings.[22 23] Some of the needs that were outlined were the need for support regarding care receiver's as well as own well-being,[23] opportunity for formalised training,[24] flexible working conditions[20] and more professional support and respite services.[25] Although it is evident that Lithuanian informal caregivers experience certain strains, more data are needed for gaining knowledge about their basic characteristics and evaluating their challenges so that the following guidelines regarding suitable support options could be proposed. More so, in the light of the COVID-19, as informal caregivers were already identified as a vulnerable group, experiencing more difficulty in providing care and increase in the burden because of the pandemic measures.[26]

Altogether, the main aim of this study was to conduct an online survey investigating Lithuanian informal caregiver characteristics, burden and general support needs as well as the association between the COVID-19 and the caregiver and care receiver well-being. More specifically, we aimed at the informal caregivers providing care in the context of disability, illness, old age or frailty. The results of this survey will be used for warranting healthcare professional, researcher, policy-maker and general public's attention towards to the Lithuanian informal caregiver support needs.

## METHOD

### Design and sample

An online survey was designed to explore characteristics, experiences and support needs of Lithuanian informal caregivers. To be eligible, participants had to be informal caregivers or have informal caregiving related experience. Also, they needed to be at least 18 years, fluent in Lithuanian language and have internet access via computer, a mobile phone, or a compatible device. We have followed The Strengthening the Reporting of Observational Studies in Epidemiology Statement: guidelines for reporting observational studies.

### Participant and public involvement

The development of the survey was inspired by the knowledge obtained following the randomised controlled trial as well as follow-up qualitative interviews with the informal caregivers. More specifically, we have identified the need to obtain more knowledge in relation to the caregiver characteristics that could be beneficial for further development and tailoring of the support for the informal caregivers.

Participants and/or the public were not directly involved in the design, or conduct, or reporting or dissemination plans of this research.

### Survey development

The survey was conducted in the Lithuanian language. The survey was developed by the authors including a researcher at Vilnius University, with survey items informed by the research questions as well as the current literature.[18 27 28] More specifically, previous research studies investigating informal caregiver and caregiving related characteristics, well-being, knowledge and support needs. An established 24-item questionnaire measuring caregiver perceived burden, the Caregiver Burden Inventory (CBI),[29] was also included. The final survey consisted of 62 multiple-choice questions. Several items also had an option for free-text answers. A short description of the survey items follows below.

### Demographic characteristics and caregiving specifics

There were 14 questions dealing with participant demographic characteristics, such as gender and education. In addition, there were 12 questions in

relation to the care receiver and caregiving in general (eg, care recipients age and gender; relation to care recipient; duration of caregiving).

### Caregiving knowledge and support needs

In this section, participants were presented with two questions in relation to their knowledge and five questions in relation to their needs and currently available as well as preferred support options (eg, What are the main caregiving related challenges that you experience; and Are you receiving caregiving related support?). Items for this section were inspired by previous research studies investigating informal caregiver use of and need for support.[27 30]

### Caregiver burden

In addition, participants were asked to fill in the CBI questionnaire.[29] The CBI consists of five subscales: Physical health, Emotional health, Time dependency, Development and Social relationships. Each of the subscales contain five questions with an exception of the Physical health, which has four questions. Response options range from 0 ('never') to 4 ('nearly always'). A total score is calculated by summing responses (range 0–96), with higher total scores indicating higher burden. Sum scores can also be calculated for the subscales separately; for Physical health (range 0–16) and the remaining four subscales (range 0–20). Overall, a score of 24 is considered to indicate a need for respite, while scores above 36—a risk of a burn-out. CBI has previously shown high reliability.[31] In this sample, reliability coefficients (Cronbach's alpha) were also found to be high: Time dependency ($\alpha=0.92$), Development ($\alpha=0.88$), Physical health ($\alpha=0.87$), Emotional health ($\alpha=0.84$) and Social relationships ($\alpha=0.82$).

### Well-being and support during COVID-19

At the end of the survey, participants were presented with three questions in relation to the COVID-19 pandemic. More specifically, participants were asked how: (1) their own well-being; (2) well-being of the care recipient and (3) availability of the support have or have not changed due to the pandemic. There were five answer options for answering the third question: I do not know; improved; did not change; worsened; worsened very much or were not available. Answer options for the first and second question were presented on a 3-point Likert scale (well-being 1-improved; 2-did not change; 3-worsened). In answering these questions, participants were also requested to indicate type of changes they observed.

### Procedure

The survey was conducted online. Data collection took part between the mid of May and the beginning of September 2020. The link to the survey was disseminated via various social media platforms. The link was also sent to some patient care organisations directly.

Interested individuals had to click on the survey link and provide informed consent before taking part in the survey.

### Statistical analysis

Data were analysed using IBM SPSS Statistics V.25. Descriptive statistics were used for summarising participant demographic and caregiving-related characteristics as well as support needs and the COVID-19 question responses. Independent samples t-tests and one-way analysis of variance (ANOVA) were performed for investigating the association between CBI scores and several demographic characteristics. Multiple linear regression was performed for selected demographic characteristics (as predictors) and CBI total score (as dependent variable). Statistical significance was set at $p<0.05$. When possible, free-text answers were categorised.

## RESULTS

### Informal caregiver demographic characteristics

A total of 226 individuals completed the survey. There were no missing data entry points as the survey could only be submitted when all questions had been answered. Demographic characteristics are presented in the table 1. As it is evident form the table, majority of the participants were women (87.6%). Given the small proportion of men participants and previously observed gender differences in caregiving prevalence and outcomes, we will report results of both genders separately.

Most of the participants had obtained a university degree (56.2%), were married or had a partner (69.5%) and were residing in the capital or one of the larger cities in Lithuania (57%). One of the more striking observations was in relation to the occupational status: after starting to provide care a substantial proportion of the participants had either started working less than full-time (from 10.6% to 17.3%) or became unemployed (from 16.8% to 38.9%). This difference was found to be statistically significant: $t(225)=-8.69$, $p<0.001$. Regarding self-perceived health, more than half of the sample indicated experiencing physical (58.8%) and psychological health (55.8%) problems.

### Caregiving specifics

Care receiver's characteristics as well as caregiving intensity-related information are presented in table 2. Most of the care receivers were women (68.1%), and the mean age was 76 years (SD=19.85). The care receiver's age varied, with the youngest being 5 years and oldest 99 years. However, only 5.8% of informal caregivers were providing care for 0–18 years old, and a majority (54%) provided care in the age range of 80–100 years old. Recipients were most commonly providing care for their parent

**Table 1** Caregiver characteristics

| Participant characteristics | Overall | Women | Men |
|---|---|---|---|
| No, n (%) | 226 | 198 (87.6) | 28 (12.4) |
| Age caregiver (year): M (SD) | 49.7 (12.7) | 49.13 (12.95) | 53.89 (10.4) |
| Residence: n (%) | | | |
| Capital or one of the larger cities | 129 (57) | 109 (55.1) | 20 (71.4) |
| Small cities or rural areas | 97 (43) | 89 (44.9) | 8 (28.6) |
| Highest education level: n (%) | | | |
| Primary education or vocational training | 7 (3.1) | 6 (3.0) | 1 (3.6) |
| Secondary education or professional qualification | 38 (16.8) | 32 (16.2) | 6 (21.4) |
| Applied science or similar | 54 (23.9) | 50 (25.2) | 4 (14.3) |
| University degree | 127 (56.2) | 110 (55.6) | 17 (60.7) |
| Marital status: n (%) | | | |
| Single | 32 (14.2) | 29 (14.6) | 3 (10.7) |
| Married/partner | 157 (69.5) | 135 (68.2) | 22 (78.6) |
| Divorced/widowed or other | 37 (16.3) | 34 (17.2) | 3 (10.7) |
| Family members: n (%) | | | |
| 1–2 | 81 (35.9) | 68 (34.4) | 13 (46.5) |
| 3–4 | 118 (52.2) | 106 (53.6) | 12 (42.8) |
| 4+ | 27 (11.9) | 24 (12) | 3 (10.7) |
| Occupational status before caregiving: n (%) | | | |
| Employed full time | 164 (72.6) | 139 (70.2) | 25 (89.3) |
| Employed part time | 24 (10.6) | 23 (11.6) | 1 (3.6) |
| Unemployed | 38 (16.8) | 36 (18.2) | 2 (7.1) |
| Occupational status after starting caregiving: n (%) | | | |
| Employed full time | 99 (43.8) | 83 (41.9) | 16 (57.1) |
| Employed part time | 39 (17.3) | 34 (17.2) | 5 (17.9) |
| Unemployed | 88 (38.9) | 81 (40.9) | 7 (25) |
| Financial situation: n (%) | | | |
| Cannot afford enough food | 42 (18.6) | 40 (20.2) | 2 (7.1) |
| Enough for food, but not for bigger purchases (eg, television) | 83 (36.7) | 70 (35.4) | 13 (46.4) |
| Enough for bigger, but not very big purchases (eg, a flat) | 92 (40.7) | 81 (40.9) | 11 (39.4) |
| Everything is affordable | 9 (4.0) | 7 (3.5) | 2 (7.1) |
| Health problems caregiver: n (yes %) | | | |
| Physical health problems | 133 (58.8) | 118 (59.6) | 15 (53.6) |
| Psychological health problems | 126 (55.8) | 115 (58.1) | 11 (39.3) |
| Self-rated well-being over last 4 weeks: n (%) | | | |
| Either very good or good | 42 (18.6) | 35 (17.6) | 7 (25) |
| Neither good nor bad | 82 (36.3) | 70 (35.4) | 12 (42.9) |
| Not very good or bad | 102 (45.1) | 93 (47) | 9 (32.1) |

(father or mother) (57.1%). Among the types of relations as categorised from the free-text answers and not mentioned in the table, the most common for the recipients was to be a grandmother (5%). Most of the care receivers had dementia (22.6%), a previous experience of stroke or myocardial infarction (15%) or needed assistance because of old age (13.7%). Regarding reasons for care provision, two of the most frequent reasons were own initiative (23%) and having no other family member available for care provision (23%). Other common reasons stated in the free-text boxes were the care receiver requesting care (6.2%) and being the parent of the care receiver (6.2%).

Almost half of the participants had provided care for more than 4 years (48.2%), 5–7 days per week (79.6%), and either 3–7 (32.7%) or more than 12 hours per day (35.4%). In addition, 8% of the participants used free-text answer option to indicate that the care receiver was fully dependent on their support.

| Table 2 Caregiving information | | | |
|---|---|---|---|
| Caregiving related information | Overall | Women | Men |
| Gender care recipient: n (%) | | 154 (68.1) | 72 (31.9) |
| Age care recipient (year): M (SD) | 71.4 (23.31) | 76.3 (19.85) | 60.9 (26.65) |
| Relation to care recipient: n (%) | | | |
| Husband/wife/partner | 23 (10.2) | 13 (6.6) | 10 (35.7) |
| Father/mother | 129 (57.1) | 116 (58.6) | 13 (46.4) |
| Parent-in-law/uncle/auntie | 23 (10.2) | 19 (9.6) | 4 (14.3) |
| Brother/sister | 5 (2.2) | 5 (2.5) | 0 |
| Daughter/son | 25 (11.1) | 25 (12.6) | 0 |
| Other | 21 (9.3) | 20 (10.1) | 1 (3.6) |
| Main reason for caregiving: n (%) | | | |
| Old age | 31 (13.7) | 27 (13.6) | 4 (14.3) |
| Dementia | 51 (22.6) | 45 (22.7) | 6 (21.5) |
| Stroke/myocardial infarction | 34 (15) | 32 (16.3) | 2 (7.1) |
| Amyotrophic lateral sclerosis | 12 (5.3) | 6 (3) | 6 (21.5) |
| Cerebral palsy | 11 (4.9) | 11 (5.6) | 0 |
| Cancer | 10 (4.4) | 8 (4) | 2 (7.1) |
| Other | 77 (34.1) | 69 (34.8) | 8 (28.5) |
| Caregiver resides with care receiver: n (yes %) | 163 (72.1) | 142 (71.7) | 21 (75) |
| Caregiving circumstances: n (single caregiver %) | 92 (40.7) | 76 (38.4) | 16 (57.1) |
| Reasons for providing care: n (%) | | | |
| Own initiative | 52 (23) | 44 (22.2) | 8 (28.6) |
| Due to close living proximity to the care receiver | 26 (11.5) | 21 (10.6) | 5 (17.9) |
| There were no other family members available for caregiving | 52 (23) | 46 (23.2) | 6 (21.4) |
| This what was agreed on together with other family members | 41 (18.1) | 39 (19.7) | 2 (7.1) |
| Other | 55 (24.4) | 48 (24.3) | 7 (25) |
| Time caring: n (in months %) | | | |
| <12 | 47 (20.8) | 45 (22.7) | 2 (7.1) |
| 12–24 | 36 (16) | 28 (14.1) | 8 (28.6) |
| 24–48 | 34 (15) | 32 (16.2) | 2 (7.1) |
| 48+ | 109 (48.2) | 93 (47) | 16 (57.1) |
| Time per week: n (in days, %) | | | |
| 1–2 | 26 (11.5) | 23 (11.6) | 3 (10.7) |
| 3–4 | 20 (8.8) | 17 (8.6) | 3 (10.7) |
| 5–7 | 180 (79.6) | 158 (79.8) | 22 (78.6) |
| Time per day: n (in hours, %) | | | |
| 3< | 51 (22.6) | 40 (20.2) | 11 (39.3) |
| 3–7 | 74 (32.7) | 64 (32.3) | 10 (35.7) |
| 8–11 | 21 (9.3) | 20 (10.1) | 1 (3.6) |
| 12+ | 80 (35.4) | 74 (37.4) | 6 (21.4) |
| Helping care receiver with basic activities of daily living (ADLs)*: n (yes, %) | | | |
| Bathing | 175 (77.4) | 160 (80.8) | 15 (53.6) |
| Brushing teeth | 102 (45.1) | 95 (48) | 7 (25) |
| Dressing | 156 (69) | 140 (70.7) | 16 (57.1) |
| Eating | 160 (70.8) | 143 (72.2) | 17 (60.7) |
| Moving | 155 (68.6) | 134 (67.7) | 21 (75) |
| Toilet needs | 130 (57.5) | 116 (58.6) | 14 (50) |
| Maintaining general hygiene (eg, cutting nails) | 189 (83.6) | 170 (85.9) | 19 (67.9) |
| Helping care receiver with instrumental ADLs*: n (yes, %) | | | |

Continued

**Table 2** Continued

| Caregiving related information | Overall | Women | Men |
|---|---|---|---|
| Using telephone | 100 (44.2) | 87 (43.9) | 13 (46.4) |
| Laundry | 167 (73.9) | 143 (72.2) | 24 (85.7) |
| Shopping | 170 (75.2) | 146 (73.7) | 24 (85.7) |
| Transportation | 158 (69.9) | 134 (67.7) | 24 (85.7) |
| Cooking | 172 (76.1) | 148 (74.7) | 24 (85.7) |
| Medication | 164 (72.6) | 147 (74.2) | 17 (60.7) |
| Household | 179 (79.2) | 152 (76.8) | 27 (96.4) |
| Financial management | 139 (61.5) | 122 (61.6) | 17 (60.7) |

*Possible to choose more than one response option.

## Informal caregiver knowledge and support needs

Several aspects in relation to the informal caregivers' knowledge and needs were identified (table 3). Almost half of the participants (47.3%) reported no specific knowledge about the disorder of the care recipient, and more than half (55.3%) reported no knowledge

**Table 3** Caregiver's knowledge and support needs

| Participant characteristics | Overall | Women | Men |
|---|---|---|---|
| No, n (%) | 226 | 198 (87.6) | 28 (12.4) |
| Prior care provision knowledge regarding the disorder(s) of care receiver: n (no knowledge, %) | 107 (47.3) | 96 (48.5) | 11 (39.3) |
| Prior care provision knowledge in general: n (no knowledge, %) | 125 (55.3) | 108 (54.5) | 17 (60.7) |
| Would you like to receive more professional support with caregiving (medical, social etc): n (yes %) | 167 (73.9) | 143 (72.2) | 24 (85.7) |
| Personal difficulties experienced by caregivers*: n (%) | | | |
| Less time for one-self | 191 (84.5) | 169 (85.4) | 22 (78.6) |
| Changes in sleep quality | 128 (56.6) | 114 (57.6) | 14 (50) |
| Changes in relationships with other people | 122 (54) | 110 (55.6) | 12 (42.9) |
| Changes in financial situation | 107 (47.3) | 95 (48) | 12 (42.9) |
| Changes in physical or psychological health | 173 (76.5) | 156 (78.8) | 17 (60.7) |
| Have you searched for caregiving related support*: n (%) | | | |
| Have not searched | 79 (35) | 67 (33.8) | 12 (42.9) |
| Yes, searched for financial support | 54 (23.9) | 45 (22.7) | 9 (32.1) |
| Yes, searched for own well-being support | 64 (28.3) | 59 (29.8) | 5 (17.9) |
| Yes, looked for professional support for conducting caregiving tasks | 78 (34.5) | 69 (34.8) | 9 (32.1) |
| Receiving caregiving related support*: n (%) | | | |
| Not receiving support | 110 (48.7) | 98 (49.5) | 12 (42.9) |
| Receiving financial support | 76 (33.6) | 68 (34.3) | 8 (28.6) |
| Receiving psychological support | 5 (2.2) | 5 (2.5) | 0 |
| Receiving professional support for caregiving tasks | 27 (11.9) | 22 (11.1) | 5 (17.9) |
| My situation would improve if*: n (%) | | | |
| I would receive psychological support | 73 (32.3) | 67 (33.8) | 6 (21.4) |
| I would receive professional caregiving related support | 102 (45.1) | 88 (44.4) | 14 (50) |
| I would receive more respite days | 100 (44.2) | 90 (45.5) | 10 (35.7) |
| I would receive financial support | 105 (46.5) | 87 (43.9) | 18 (64.3) |
| I would receive more information about the care provision and specific disorder | 59 (26.1) | 53 (26.8) | 10 (35.7) |
| I would receive more support from people in my close environment | 59 (26.1) | 56 (28.3) | 3 (10.7) |
| If time spent caregiving would add to the years of working | 125 (55.3) | 109 (55.1) | 16 (57.1) |

*Possible to choose more than one response option.

about how to provide care in general. Consequently, a majority wished to receive more professional support in their role as caregivers (73.9%). Less time for oneself and changes in physical and mental health were identified as the two most prominent challenges, 84.5% and 76.5%, respectively. Regarding support, most of participants either did (34.5%) or did not look for professional support (35%). In turn, almost half of the participants reported that they were not receiving any caregiving-related support (48.7%). Regarding the ones receiving support, financial aid was the most mentioned (33.6%). Only 2.2% of the participants received psychological help. Participants reported that their situation would improve if caregiving was recognised as part of working experience (55.3%), if they would receive financial (46.5%) or professional support (45.1%), and more days for respite (44.2%).

### Informal caregiver burden

Mean scores, SDs and gender differences regarding scores on the CBI and the separate subscales are presented in table 4. Since all the subscales have five items each except for the Physical health subscale, which has four, the scores for this subscale were multiplied by 1.25.[29] Overall, participants displayed high mean score on the CBI (M=50.21, SD=15.63), with highest mean score on the subscale Time dependency (M=16.15, SD=4.21). As illustrated in table 4, women scored significantly higher on the overall CBI score (p=0.011) as well as on the subscales development (p=0.035) and Physical health (p=0.002).

Independent samples t-tests or ANOVAs were performed when analysing demographic as well as caregiving-related characteristics (with exception of the care receiver symptoms, which was not included due to the many categories) in relation to CBI total scores (table 5). Eight variables (nine if gender is included) were found to be associated with increased CBI scores: physical (p<0.001) or psychological (p<0.001) health complaints, poorer self-rated well-being (p<0.001), residing with the care receiver (p<0.001) and caring for longer and with higher intensity (p<0.001). Also, informal caregivers who started

providing care as there were no other family members to help were found to be significantly more burdened than individuals who took up this task following own initiative (p=0.001).

We ran a hierarchical multiple linear regression including the significant predictors presented in table 5. Out of the nine entered predictors, four made a significant independent contribution to CBI total score: self-rated well-being (p=0.001), caregiving duration in months (p=0.006), caregiver's gender (p=0.046) and experience of the psychological health problems (p=0.001) (table 6, block 1). We ran the regression again with all four variables included. All variables were found to contribute to the model significantly, explaining 27.3% of variance in the CBI scores (table 6, block 2).

### Well-being during COVID-19

Out of the 226 participants, a majority indicated that neither their own (63.7%) nor the care receiver's well-being (68.1%) had changed during the COVID-19 pandemic. Regarding the availability of the care-related support, 27.4% indicated a decrease and 32.3% a very big decrease in the availability of support.

## DISCUSSION

The aim of this study was to investigate Lithuanian informal caregiver characteristics, support needs, burden and the impact of the COVID-19 on the well-being. Overall, informal caregivers in this survey displayed high burden, high involvement in the care provision and limited access to the support options. Most of the participants indicated no changes in their well-being due to the COVID-19. We further discuss the findings as well as the limitations of this study below.

### Caregiver and caregiving-related characteristics

The mean age of the participants in the survey (M=49) could be deemed compatible with median age of Lithuanian citizens (M=45)[32] and is in line with the research literature indicating that most of the informal care in Lithuania, as in other parts of the world, is carried out

**Table 4** Means, SDs and independent samples t-test results for CBI total score and separate subscales

| Scale | Mean (SD) | | | t | P value |
| --- | --- | --- | --- | --- | --- |
| | Overall | Women | Men | | |
| CBI total score | 50.21 (15.63) | 51.2 (15.41) | 43.18 (15.68) | −2.57 | 0.011 |
| Time dependency | 16.15 (4.21) | 16.30 (4.16) | 15.07 (4.48) | −1.45 | 0.149 |
| Development | 12.77 (4.85) | 13.03 (4.86) | 10.96 (4.44) | −2.12 | 0.035 |
| Physical health | 11.07 (4.85) | 11.44 (4.76) | 8.44 (4.75) | −3.12 | 0.002 |
| Emotional health | 4.93 (4.04) | 5.05 (4.04) | 4.11 (4.0) | −1.15 | 0.251 |
| Social relationships | 7.5 (4.71) | 7.68 (4.58) | 6.29 (5.45) | −1.47 | 0.144 |

CBI, Caregiver Burden Inventory.

**Table 5** Caregiver burden associations with sociodemographic and informal caregiver study variables

| Variable | CBI, M (SD) | T or F* | P value |
|---|---|---|---|
| Age caregiver | | | |
| 18–39 | 47.07 (2.15) | 1.61 | 0.203 |
| 40–59 | 51.66 (15.86) | | |
| 60–80 | 48.98 (16.09) | | |
| Residence | | | |
| Capital or one of the larger cities | 48.9 (17.31) | −1.51 | 0.131 |
| Small cities or rural areas | 51.95 (12.96) | | |
| Education | | | |
| Primary education or vocational training | 49.71 (15.91) | 0.78 | 0.504 |
| Secondary education or professional qualification | 48.76 (12.52) | | |
| Applied science or similar | 53.02 (14.89) | | |
| University degree | 49.47 (16.75) | | |
| Marital status | | | |
| Single | 49.09 (12.57) | 0.15 | 0.862 |
| Married/partner | 50.21 (15.56) | | |
| Divorced/widowed or other | 51.16 (18.44) | | |
| Family members | | | |
| 1–2 | 49.09 (15.37) | 1.93 | 0.148 |
| 3–4 | 51.94 (15.75) | | |
| 4+ | 46.0 (15.33) | | |
| Financial situation | | | |
| Cannot afford enough food | 52.67 (18.82) | 0.42 | 0.737 |
| Enough for food, but not for bigger purchases (eg, television) | 49.7 (13.60) | | |
| Enough for bigger, but not very big purchases (eg, a flat) | 49.61 (15.79) | | |
| Everything is affordable | 49.56 (16.95) | | |
| Current occupational status | | | |
| Employed full time | 48.52 (16.07) | 1.05 | 0.352 |
| Employed part time | 51.90 (17.84) | | |
| Unemployed | 51.36 (14.01) | | |
| Physical health problems caregiver | | | |
| No | 45.82 (15.74) | −3.59* | <0.001 |
| Yes | 53.28 (14.86) | | |
| Psychological health problems caregiver | | | |
| No | 43.8 (14.83) | −5.86 | <0.001 |
| Yes | 55.29 (14.38) | | |
| Self-rated well-being over last 4 weeks | | | |
| Either very good or good | 36.88 (15.18) | 22.88 | <0.001 |
| Neither good nor bad | 52.09 (12.36) | | |
| Not very good or bad | 54.19 (15.35) | | |
| Gender care receiver | | | |
| Men | 49.04 (15.53) | −0.77 | 0.444 |
| Women | 50.75 (16.14) | | |
| Age care receiver | | | |
| 0–18 | 53.31 (5.22) | 1.14 | 0.337 |
| 19–39 | 54.59 (14.62) | | |
| 40–59 | 44.45 (8.89) | | |
| 60–79 | 50.19 (17.43) | | |

Continued

**Table 5** Continued

| Variable | CBI, M (SD) | T or F* | P value |
|---|---|---|---|
| 80–100 | 50.22 (16.39) | | |
| Relation care receiver | | | |
| Husband/wife/partner | 45.39 (13.71) | 1.72 | 0.132 |
| Father/mother | 51.19 (15.87) | | |
| Parent-in-law/uncle/auntie | 46.87 (20.09) | | |
| Brother/sister | 49.0 (10.56) | | |
| Daughter/son | 56.04 (8.24) | | |
| Other | 46.48 (16.70) | | |
| Individual is the only caregiver | | | |
| Yes | 52.37 (14.30) | 1.73 | 0.085 |
| No | 48.72 (16.37) | | |
| Reason why one started providing care | | | |
| Own initiative | 44.17 (16.50) | 4.59 | 0.001 |
| Due to the proximity to the care receiver | 47.31 (19.70) | | |
| No other family members available | 56.46 (13.17) | | |
| Decided with family members | 50.27 (15.32) | | |
| Other | 51.33 (12.88) | | |
| Receiving of caregiving related support | | | |
| Receiving support | 48.89 (16.99) | 1.31 | 0.191 |
| Not receiving support | 51.6 (14.0) | | |
| Residing with the care receiver | | | |
| Yes | 52.9 (13.28) | 8.42 | <0.001 |
| No | 43.24 (18.91) | | |
| Time caring: months | | | |
| <12 | 45.17 (19.20) | 7.99 | <0.001 |
| 12–24 | 42.56 (14.72) | | |
| 24–48 | 51.56 (15.44) | | |
| 48+ | 54.49 (12.62) | | |
| Time week: days | | | |
| 1–2 | 39.04 (19.87) | 8.12 | <0.001 |
| 3–4 | 49.90 (18.14) | | |
| 5–7 | 51.86 (14.01) | | |
| Time day: hours | | | |
| <3 | 39.90 (19.07) | 11.82 | <0.001 |
| 3–7 | 52.12 (13.87) | | |
| 8–11 | 50.0 (9.85) | | |
| 12+ | 55.06 (13.04) | | |

*Either Independent samples t-tests or one-way analysis of variance were performed dependently on the number of categories.
CBI, Caregiver Burden Inventory.

by women in their fifties.[6] More than half of the participants reported either psychological or physical health problems suggesting that caregivers are at risk of poor well-being.[10 11 19] We also observed a significant decrease in the proportion of caregivers working full time after the start of the care provision. Current labour market-related measures for informal caregivers in Lithuania could be described as limited and insufficient.[20] This might explain why participants in our study had to move from working full-time to either not working at all or working reduced number of hours. Having to reduce work hours due to the caregiving duties alone has previously been found to have a negative effect on the caregiver's psychological well-being.[33] Further efforts to improve current labour market-related measures are most likely to be needed to prevent such risks.

Individuals who provide care for 11 or more hours per week have previously been defined as intensive caregivers.[6]

**Table 6** Multiple linear regression results with demographic characteristics as predictors and CBI as an outcome variable

| | Block 1 | | | | |
|---|---|---|---|---|---|
| Included variables | B (95% CI) | SE B | β | P value | $R^2$ |
| Self-rated well-being over last 4 weeks | 4.05 (1.77 to 6.32) | 1.15 | 0.23 | 0.001 | 0.324 |
| Time day: hours | 0.94 (−0.78 to 2.65) | 0.87 | 0.07 | 0.283 | |
| Time week: days | 2.63 (−0.62 to 5.89) | 1.65 | 0.11 | 0.112 | |
| Time caring: months | 2.17 (0.64 to 3.70) | 0.78 | 0.17 | 0.006 | |
| Reason why one started providing care | 0.87 (−0.32 to 2.06) | 0.60 | 0.08 | 0.151 | |
| Residing with the care receiver | 2.46 (−2.62 to 7.54) | 2.58 | 0.07 | 0.341 | |
| Gender caregiver | 5.50 (0.09 to 10.90) | 2.74 | 0.12 | 0.046 | |
| Physical health problems caregiver | 1.43 (−2.46 to 5.32) | 1.97 | 0.05 | 0.469 | |
| Psychological health problems caregiver | 6.33 (2.48 to 10.18) | 1.95 | 0.20 | 0.001 | |
| | Block 2 | | | | |
| Included variables | B (95% CI) | SE B | β | P value | $R^2$ |
| Self-rated well-being over last 4 weeks | 4.79 (2.64 to 6.94) | 1.09 | 0.27 | <0.001 | 0.273 |
| Time caring: months | 3.02 (1.54 to 4.50) | 0.75 | 0.23 | <0.001 | |
| Gender caregiver | 6.26 (0.83 to 11.69) | 2.76 | 0.13 | 0.024 | |
| Psychological health problems caregiver | 6.77 (2.90 to 10.63) | 1.96 | 0.22 | 0.001 | |

CBI, Caregiver Burden Inventory.

In our sample, 77.4% of all the participants fell into this category. Most of these caregivers provided care for 5–7 days per week, and at the time of the survey completion, for 4 or more years. Mental health consequences have previously been found to be even more severe for intensive caregivers,[6] a finding that could at least partly explain the sample's high overall scores on the CBI measure. In line with this, most of the participants indicated that they would like to receive more professional support. In terms of available support, current day care and nursing home services as well as respite services for the informal caregivers in Lithuania could be described as very limited.[20] This suggests that further policy measures for improving both, availability and accessibility of such services are needed.

### Caregiver knowledge and support needs

Most of the participants started providing care without having any general or receiver symptom-specific knowledge about caregiving. Informal caregivers in Lithuania often have to learn about the care provision through own experience.[24] As a consequence, they might experience feelings of anxiety and uncertainty. In addition, almost half of the participants did not receive any support in their caregiving. Among those who had support, a majority received financial support. Time spent for caregiving being counted as work experience was the most favoured suggestion by the caregivers. In addition, a majority indicated that financial and professional support would improve their situation. Interestingly, approximately one-third of the participants indicated that they had not searched for support. One explanation could be that they did not know which support is available or how

to access it.[24] Prior negative experiences of interactions with healthcare professionals could also influence healthcare seeking. A recent qualitative study on Lithuanian informal caregivers reported that some caregivers experienced difficulty in communicating with the healthcare professionals.[22] Studies in other countries have also found that carers experience dissatisfaction with the healthcare providers in terms of information provision, treatment optimisation, involvement of the caregiver and management of caregivers' own health.[34] As a solution, additional training could be offered to the professionals about guiding and supporting informal caregivers.[35] Early initiation of the contact with the caregivers could also be useful. This might be especially important for cases in which help-seeking behaviour conflicts with caregivers' values[36] or caregivers express high needs for continuous or frequent support.

### Caregiver burden

In line with the previous literature,[4 6] we found that women participants experienced a higher burden than the men. Participants overall scored the highest on the Time dependency subscale of the CBI which mirrors a large time investment on caregiving duties. This was further outlined by the regression analyses, in which being a woman, longer caregiving duration, poorer self-rated well-being and psychological health problems were significant predictors of higher CBI total scores. The question of what type of psychological support options informal caregivers would prefer remains. As identified in the recent qualitative Lithuanian informal caregiver study[24] access to peer support groups as well as internet-based intervention

programmes could have potential in reducing caregiver psychological burden. Further research into these matters is encouraged.

## COVID-19

Contrary to our expectations and recent researcher findings,[26 37–39] most of the informal caregivers did not report any changes in own or care receivers well-being because of the COVID-19 pandemic. One possible explanation could stem from the finding that in comparison to other European countries, in Lithuania, comparably lower number of cases as well as COVID-19-related deaths were reported during the first wave of the pandemic.[40] Lithuanian government has also taken early preventative measures which were deemed as innovative and promising in bettering the social policies.[41] Alternatively, higher appreciation of the life at the start of the pandemic could be another explanation why no changes were observed.[42] As outlined recently, changes in the caregiver burden during the pandemic might be rather complex and vary by gender.[43] Therefore, it is possible that our questions did not capture the complexity of such changes. Future studies are needed to evaluate the impact of the pandemic on informal caregivers.

## Study limitations

There are limitations to be addressed. First, our sample might not be representative of Lithuanian caregivers as participants are likely to have higher computer literacy and motivation to participate in online research. Even though internet access is widely spread throughout the country, people in their fifties were found to access the internet less often than the younger age groups.[44] In addition, submission of the survey responses was only possible on completion of all items. This could have had an influence on participant motivation to complete the survey and hence, add to the sample selection bias. Second, caregiver knowledge and support needs as well as changes in well-being during the COVID-19 were investigated by the use of only few items. Therefore, findings should be replicated using established and validated questionnaires. Third, our study was cross-sectional and did only investigate caregiver needs at a certain point in time. Longitudinal data should be collected for more comprehensive evaluation of the possible fluctuations in well-being and support needs over time. Also, for the COVID-19, considering the changing nature of the pandemic. Finally, it must be emphasised that due to the cross-sectional design of this research study, all findings are descriptive, indicating that no causal inferences should be drawn.

## CONCLUSION AND RECOMMENDATIONS

Based on the results of our sample we conclude that the Lithuanian informal caregivers, in relation to caregivers in other European countries, experience high burden and unmet practical as well as psychological support needs. We outline here a few points that could be focused on by policy-makers, healthcare professionals and researchers. First, current labour market policies are insufficient in allowing caregivers to balance caregiving, work and personal life. To prevent possible negative financial and psychological health consequences for the caregivers, further emphasis should be put on adapting current policies. Second, we found the caregivers to express the need for information and practical support. More accessible information sources and better guidance from health professionals could be offered. Lastly, participants in our study were found to experience high caregiver burden. Due to the low coverage and accessibility of psychological support options, we encourage researchers to develop innovative support measures, such as online support groups or psychological support interventions.[45]

We conclude that supporting informal caregivers is crucial not only for the individual, but also on a societal level. Meeting these needs is important from the start and throughout the caregiving experience.

**Acknowledgements** We thank George Vlaescu and Austeja Dumarkaite for the help with the study. We also thank all the individuals and organisations that willingly helped with the dissemination of the survey.

**Contributors** All authors contributed to the conception and study design of the study. IB and EK contributed to the data collection. IB and GA analysed and interpreted the data. IB drafted the manuscript. EK, RS and GA critically revised the paper. GA is the guarantor. All authors approved the final version of the manuscript.

**Funding** This project has received funding from the European Union's Horizon 2020 research and innovation programme under the Marie Skłodowska-Curie grant agreement No 814072 and is part of The European Training Network on Informal Care (ENTWINE).

**Competing interests** None declared.

**Patient and public involvement** Patients and/or the public were not involved in the design, or conduct, or reporting, or dissemination plans of this research.

**Patient consent for publication** Not applicable.

**Ethics approval** Ethics approval for the study was granted by the Vilnius University Psychology Research Ethics Committee, 08-07-2019 No.26.

**Provenance and peer review** Not commissioned; externally peer reviewed.

**Data availability statement** Data are available on reasonable request. Anonymised data will be stored at Linköping university for 10 years. It will be available on reasonable request in excel format after the publication of the manuscript. Primary investigator (GA, Linköping University; gerhard.andersson@liu.se) should be contacted for requesting about the data.

**ORCID iDs**
Ieva Biliunaite http://orcid.org/0000-0002-9111-0891
Evaldas Kazlauskas http://orcid.org/0000-0002-6654-6220

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
