## [Reviewer comments · BMJ Open]

ARTICLE DETAILS

TITLE (PROVISIONAL)	Informal caregiver support needs and burden: a survey in Lithuania
AUTHORS	Biliunaite, Ieva; Kazlauskas, Evaldas; Sanderman, Robbert; Andersson, Gerhard

VERSION 1 – REVIEW

REVIEWER	Ballesteros, Javier Universidad del Pais Vasco, Neuroscience
REVIEW RETURNED	01-Aug-2021

GENERAL COMMENTS	This manuscript presents the burden and support needs of informal caregivers in one of the European areas that is growing faster in the ageing population. It is a cross-sectional study with a survey designed to be answered by internet. The highly likely selection bias of respondents is appropriately discussed and presented as a limitation. However there is also another limitation not so thoroughly discussed and is that the survey does not present missing values in any variable since non-answered questions were not allowed by the system. I think it adds to the sample selection bias and must be also discussed as limitation. The only other point I have is to discuss the results showed on table 6 as results coming from an exploratory approach since some readers could interpret the reported associations from the hierarchical models as causal.
--

REVIEWER	Vilchinsky, Noa Universitat Bar-Ilan, Department of Psychology
REVIEW RETURNED	15-Sep-2021

GENERAL COMMENTS	Dear Editor Thank you for the opportunity to review the manuscript titled: "Informal caregiver support needs and burden: a survey in Lithuania" The manuscript presents a survey study in Lithuania aiming at mapping caregivers' needs and characteristics. I commend the authors for focusing on a country which is much less studied compared to "Westernized" countries. This bares an important contribution to the literature. Also, observing the support needs of informal caregivers is crucial in our times, especially in the face of the current global pandemic. Thus, the study aim is important and of value. Overall, the manuscript is very interesting, and the authors raise important themes that can direct future interventions. I wish to point a few issues that in my opinion require amendment before considering acceptance.
---

	My detailed comments are as follow:  1. First, the manuscript must be edited by a professional English editor. 2. The abstract should include not only the aim of the study but also the rationale for the study as well as it's importance. 3. Page 4, line 5: the authors should better explain that they refer to informal caregiving in times of illness, disability, frailty, or old age and not merely ordinary familial care. 4. Since this is a cross sectional retrospective study, the authors should avoid wording insinuating causality as: (page 5, line 6): "the effects of the COVID-19 on the caregiver and care-receiver well-being" (this is only one example, there are many more). 5. I believe the readers would benefit for more information about the Lithuanian culture. In what ways is it a more traditional than West-European countries? What are the common attitudes regarding the need to take care of a family member? Do people tend to live with their elderly? What is the economic situation in this country and are there any pro- caregivers' specific legislations or benefits? 6. In the same vein, few more details on the COVID-19 situation in Lithuania is needed. Has this country suffered badly compared to others in the EU, for example? 7. Also, how frequent is the use of online applications in Lithuania? This is relevant for the question of generalizability as only internet-literate people could have participate. 8. Overall, it will be helpful if the authors could provide some more information about the Lithuanian population so that the readers could infer about the representativeness of the sample as compared to the population in large. For example, it seems that the sample is comprised of young people on average (M= 49). Is that compatible with the mean age in Lithuania? If not, it deserves a specific discussion. 9. Page 5, line 25: the authors are advised to add information about the: " randomized controlled trial as well as follow-up qualitative interviews with the informal caregivers" which guided their current study. 10. The authors are advised to provide more information about the scales used to measure participants knowledge and needs. 11. Page 5, line 34: It will be helpful for the readers if the authors could elaborate a bit more on the literature on which they have based their choice of items. 12. Gender and age are presented on both Tables 1 and 2. I suggest presenting these data only on one table for parsimony. 13. The discussion is well integrated and comprehensive. Yet, the authors should be cautious and remind the readers that their conclusions are based on a non-representative sample. 14. The second point of strengths and limitations was not clear to me It. For example, it is not very clear how further studies will: "meet the growth of the demand for informal care in the future". I suggest re wording for better clarity. 15. Third point of strengths and limitations - since this is a retrospective cross- sectional study, the authors cannot infer "valuable insights into the changes of informal caregivers' wellbeing." I suggest toning down this statement. 16. I prefer using the terms "men and women" rather than "male and females" since we are focusing on human participants.
--	--

VERSION 1 – AUTHOR RESPONSE

REVIEWER 1:

This manuscript presents the burden and support needs of informal caregivers in one of the European areas that is growing faster in the ageing population. It is a cross-sectional study with a survey designed to be answered by internet. The highly likely selection bias of respondents is appropriately discussed and presented as a limitation.

1. However there is also another limitation not so thoroughly discussed and is that the survey does not present missing values in any variable since non-answered questions were not allowed by the system. I think it adds to the sample selection bias and must be also discussed as limitation.

➤ Thank you for this valuable remark. We have now addressed this in the study limitation section.

2. The only other point I have is to discuss the results showed on table 6 as results coming from an exploratory approach since some readers could interpret the reported associations from the hierarchical models as causal.

➤ To address your remark, we have now outlined this aspect in the study limitation section. In addition, we have proofread the text to check that the wording is appropriate and does not mislead the reader to imply causality.

REVIEWER: 2

Dear Editor, Thank you for the opportunity to review the manuscript titled:

"Informal caregiver support needs and burden: a survey in Lithuania". The manuscript presents a survey study in

Lithuania aiming at mapping caregivers' needs and characteristics. I commend the authors for focusing on a country which is much less studied compared to "Westernized" countries. This bears an important contribution to the literature. Also, observing the support needs of informal caregivers is crucial in our times, especially in the face of the current global pandemic. Thus, the study aim is important and of value. Overall, the manuscript is very interesting, and the authors raise important themes that can direct future interventions. I wish to point a few issues that in my opinion require amendment before considering acceptance.

My detailed comments are as follow:

1. First, the manuscript must be edited by a professional English editor.

➤ Thank you for your valuable remark. We have now proofread the text.

2. The abstract should include not only the aim of the study but also the rationale for the study as well as its importance.

➤ Thank you for your suggestion. To improve the abstract, we have now followed your suggestion and included short introduction. In addition, we would like to outline that we had to remain restrictive in expanding further due to the maximum of the allowed word count.

3. Page 4, line 5:

the authors should better explain that they refer to informal caregiving in times of illness, disability, frailty, or old age and not merely ordinary familial care.

➤ Thank you, we have now clarified this accordingly: 'More specifically, we aimed at the informal caregivers providing care in the context of disability, illness, old age, or frailty.'

4. Since this is a cross-sectional retrospective study, the authors should avoid wording insinuating causality as: (page 5, line 6): "the effects of the COVID-19 on the caregiver and care-receiver well-being" (this is only one example, there are many more).

➤ Thank you, we have rephrased this statement accordingly. In addition, to address your remark, we have proofread the text and included an additional statement in the study limitations section.

5. I believe the readers would benefit from more information about the Lithuanian culture. In what ways is it more traditional than West-European countries? What are the common attitudes regarding the need to take care of a family member? Do people tend to live with their elderly? What is the economic situation in this country and are there any pro-caregivers' specific legislations or benefits?

➤ Thank you for your remarks. We have now touched upon these aspects in the introduction. We must outline, that due to the limitations in the scope (suggested maximum of 4000), we had discussed these aspects in a condensed manner.

6. In the same vein, few more details on the COVID-19 situation in Lithuania are needed. Has this country suffered badly compared to others in the EU, for example?

➤ We have now addressed this in the discussion section of the article.

7. Also, how frequent is the use of online applications in Lithuania? This is relevant for the question of generalizability as only internet-literate people could have participated.

➤ We have now addressed this in the study limitations section.

8. Overall,

it will be helpful if the authors could provide some more information about the Lithuanian population so that the readers could infer about the representativeness of the sample as compared to the population in large. For example,

it seems that the sample is comprised of young people on average ($M = 49$).

Is that compatible with the mean age in Lithuania? If not, it deserves a specific discussion.

➤ Thank you for your suggestion. We have now shortly addressed this in the discussion section.

9. Page 5, line 25: the authors are advised to add information about the: "randomized controlled trial as well as follow-

up qualitative interviews with the informal caregivers" which guided their current study.

➤ We have now added an additional statement explaining how our previous work guided the establishment of the survey. We must outline, that to comply with the journal suggestions, we aimed to keep this section concise and clear and hence, we did not elaborate on our previously conducted studies.

10. The authors are advised to provide more information about the scales used to measure participants' knowledge and needs.

➤ We have now added references of the work that has inspired our research items for these outcomes. In addition, since we did not use a validated scale, we have also reflected upon this in the limitations section.

11. Page 5, line 34: It will be helpful for the readers if the authors could elaborate a bit more on the literature on which they have based their choice of items.

➤ To address your remark we have now included a short explanation and a couple of examples of references to the literature.

12. Gender and age are presented on both Tables 1 and 2.

I suggest presenting these data in only one table for parsimony.

➤ Thank you for your suggestion. We would like to clarify that the Table 1 presents caregiver characteristics, while the Table 2 – care-receiver. Removing this information from either of the tables would result in missing valuable characteristics. Due to this reason, we have decided to keep this information in both tables.

13.The discussion is well integrated and comprehensive. Yet, the authors should be cautious and remind the readers that their conclusions are based on a non-representative sample.

➤ Thank you for this remark. We have now emphasized that the findings are in relation to the collected sample. In addition, we have also expanded on the sampling bias in this study.

14.The second point of strengths and limitations was not clear to me It. For example, it is not very clear how further studies will: "meet the growth of the demand for informal care in the future". I suggest re wording for better clarity.

➤ Thank you for your observation. Following your (as well as the comments from the editor), we have now completely re-worked this section.

15.Third point of strengths and limitations - since this is a retrospective cross- sectional study, the authors cannot infer "valuable insights into the changes of informal caregivers' wellbeing." I suggest toning down this statement.

➤ Please kindly refer to our previous response.

16.I prefer using the terms "men and women" rather than "male and females" since we are focusing on human participants.

➤ We have now replaced "male and females" with "men and women".

VERSION 2 – REVIEW

REVIEWER	Ballesteros, Javier Universidad del Pais Vasco, Neuroscience
REVIEW RETURNED	29-Nov-2021

GENERAL COMMENTS	I think most of the previous comments have been addressed by the authors. I would be happier with an advert regarding p-values not corrected for multiple comparisons and to stress of the exploratory characteristics of the paper, but I think the current version fits well with the requirements I asked for.
---

REVIEWER	Vilchinsky, Noa Universitat Bar-Ilan, Department of Psychology
REVIEW RETURNED	08-Dec-2021

GENERAL COMMENTS	I commend the authors for addressing all my comments in a full manner.
--